# Cardiovascular Magnetic Resonance Imaging as an Adjunct to the Evaluation of Cardiovascular Involvement in Diabetes Mellitus

**DOI:** 10.3390/jpm13050724

**Published:** 2023-04-25

**Authors:** Sophie I. Mavrogeni, George Markousis-Mavrogenis, Flora Bacopoulou, George P. Chrousos

**Affiliations:** 1Onassis Cardiac Surgery Center, 17674 Athens, Greece; 2University Research Institute of Maternal and Child Health and Precision Medicine, Medical School, National and Kapodistrian University of Athens, Aghia Sophia Children’s Hospital, 11527 Athens, Greece; 3Center for Adolescent Medicine and UNESCO Chair in Adolescent Health Care, Medical School, National and Kapodistrian University of Athens, Aghia Sophia Children’s Hospital, 11527 Athens, Greece

**Keywords:** diabetes mellitus, cardiovascular, imaging, cardiovascular magnetic resonance

## Abstract

Diabetes mellitus (DM) is a new epidemic which has presented an immense increase in recent decades, due to the rapid increase in obesity. Cardiovascular disease (CVD) significantly reduces life expectancy and is the main cause of death in type 2 diabetes mellitus (T2DM). Strict glycemic control is a well-established method to combat microvascular CVD of type 1 diabetes mellitus (T1DM); its role against CVD of the T2DM risk has not been well documented. Therefore, the most efficient prevention is multifactorial risk factor reduction. Recently, the European Society of Cardiology published its 2019 recommendations on CVD in DM. Although all clinical points were discussed in this document, only a few comments were presented about when and how we should recommend cardiovascular (CV) imaging. Currently, CV imaging is the *“must”* in CV noninvasive evaluation. Alterations in CV imaging parameters can lead to early recognition of various types of CVD. In this paper, we briefly discuss the role of noninvasive imaging modalities, emphasizing the benefits of including cardiovascular magnetic resonance (CMR) in the evaluation of DM. CMR, in the same examination, can provide an assessment of tissue characterization, perfusion and function, with excellent reproducibility and without radiation or limitations, due to the body habitus. Therefore, it can play a dominant role in the prevention and risk stratification of DM. The suggested protocol for DM evaluation should include routine annual echocardiographic evaluation of all DM patients and CMR assessment of those with poorly controlled DM, microalbuminuria, heart failure, arrhythmia and recent alterations in clinical or echocardiographic evaluation.

## 1. Introduction

Diabetes mellitus (DM) is a new epidemic which has presented an immense increase in recent decades due to the exponential increase in obesity. According to the International Diabetes Federation (IDF), about 415 million people worldwide have DM, of whom 91% have type 2 diabetes mellitus (T2DM) [1], and the number of DM patients is predicted to rise to 642 million by the year 2040 [1]. Furthermore, based on the Framingham Heart Study data, Abraham et al. [2] showed that overall annual incidence rates of DM/1000 persons rose from 3 in the 1970s to 5.5 in the first decade of the 2000s. This increment represents an 83.3% rise in the incidence of T2DM and was higher for males than females by 1.61.

Cardiovascular disease (CVD) is the main cause of death in patients with T2DM, especially in low-/middle-income countries. Universally, CVD affects about 32.2% of all patients with T2DM and represents a significant cause of mortality, as it accounts for approximately 50% of all deaths over the period 2007–2017, with stroke and coronary artery disease (CAD) being the main causes [3]. T2DM can decrease life expectancy by 10 years, with CVD being the major cause of death [1,4]. Furthermore, death rates due to CVD over seven years, in patients with or without T2DM, were 15.4% for those with no history of myocardial infarction (MI) and 42.0% for those with a prior MI. On the other hand, death rates due to CVD, in those without T2DM, were 2.1 and 15.9%, respectively [5]. Recently, a decline in CVD/mortality was noticed in diabetic and non-diabetic patients, which is steeper in the former than in the latter patients. However, despite the parallel decline, a two-fold difference in CVD/mortality between the two populations exists. A better glycemic control and CV risk factors management can potentially improve the outcome of DM. However, a rising prevalence of DM will continue to increase the absolute disease burden worldwide [4].

The high clinical impact of CVD in patients with T2DM underlines the importance of the joint management of CVD and T2DM, with strict glycemic control remaining the main focus of treatment. The role of strict glycemic control against CVD and microvascular complications has been well documented in T1DM [6,7]; however, its value in reducing the risk of CVD is not yet clearly documented in patients with T2DM [8,9,10]. Therefore, the multifactorial risk factor control is considered the most efficient prevention of macrovascular complications, which includes exercise, diet, normalization of glycemic levels, dyslipidemia treatment, cessation of smoking, and control of blood pressure.

With this aim in mind, the European Society of Cardiology (ESC) recently published its 2019 recommendations on “what we have to do in diabetic patients” [11]. According to these recommendations, (a) microalbuminuria should be assessed on a routine basis to identify those patients at risk of renal impairment and/or CVD, (b) patients with DM and hypertension, or suspected CVD, should undergo a resting electrocardiogram (ECG), (c) in patients with moderate or high risk of CVD, other tests, i.e., an ankle–brachial index (ABI), a coronary artery calcium (CAC) score, and transthoracic echocardiography can be performed to assess underlying heart disease or as risk modifiers, and (d) an assessment of novel biomarkers on a routine basis is not warranted for CV risk stratification [11]. Evaluating these recommendations, we noticed that although many points regarding this topic were discussed in detail, it was not clear when and how we should recommend CV imaging. Currently, it is well established that CV imaging represents a *“must”* in clinical CV evaluation. Alterations in CV imaging parameters may lead to early identification of various types of CVD. Furthermore, CV imaging provides direct information regarding the status of both nyocardium and coronary arteries that cannot be reached by blood biomarkers. The main CV imaging modalities currently used in the evaluation of DM include.

## 2. Echocardiography (Echo)

Echocardiography represents the main imaging modality, used in the clinical practice of cardiology, because it is a cost-effective, widely available, bedside modality that lacks ionizing radiation. It provides reliable data on the function of both the atrium and ventricles of the heart and can reveal early diastolic dysfunction usually found in early DM [12]. Two-dimensional Doppler and speckle tracking echocardiography facilitates the accurate assessment of cardiac function, valvular integrity, peri-myocardial status and cardiac hemodynamics. This information is necessary for risk stratification concerning CVD in diabetic patients and for treatment decisions [13]. Furthermore, stress echocardiography and assessment of the coronary flow reserve provide valuable information regarding the response of the heart during stress [14,15]. More specifically, echocardiography should be recommended as part of the routine annual evaluation in all diabetic and pre-diabetic patients, because it is a bedside, cost-effective, non-invasive, and easily applicable imaging modality.

The most common echo finding in both symptomatic and asymptomatic DM patients is diastolic dysfunction. According to a recent publication, the aggravation of left ventricle (LV) diastolic function is more evident in DM patients than controls. Furthermore, the probability of LV diastolic dysfunction is higher with greater HbA1c and a longer duration of DM [16].

Recently, speckle tracking echocardiography detected subtle myocardial alterations in DM patients before they develop symptoms and before conventional echocardiography detects any change. Global longitudinal strain (GLS), assessed by speckle tracking echocardiography (STE), was impaired in patients with DM, compared to controls [17]. Furthermore, a study of 230 asymptomatic patients with left ventricle ejection fraction (LVEF) ≥50% and T2DM found that approximately half of them suffered subclinical systolic impairment, as shown in GLS reduction [18]. GLS, in a 10-year follow-up, has been independently associated with all-cause hospitalization and mortality, with an increase of 10% in relative risk per 1% decrease in GLS. Furthermore, there is a prognostic significance in the recognition of epicardial CAD by stress echocardiography [19,20]. Given the increasing evidence of coronary microvascular disease in both T1DM and T2DM, stress echocardiography can be helpful in identifying this pathology. Although there are contradictory data, a recent study has also supported the theory that GLS, during a dobutamine stress echo test (DSE), can identify early subtle alterations in asymptomatic patients with T2DM vs. controls [21]. Lastly, the stress echocardiographic measurement of the coronary flow reserve (CFR), which reflects microvascular disease in the absence of flow-limiting epicardial coronary artery disease, is related to major adverse CV events in patients with T2DM, angina and non-obstructed coronary arteries evident by invasive angiography [22]. In addition, a diminished CFR in the absence of resting wall motion abnormalities has been related to adverse outcomes in asymptomatic patients [23]. However, echocardiography has serious limitations, including operator and patient body habitus dependency, low reproducibility and lack of tissue characterization [24].

## 3. Nuclear Techniques (SPECT-PET)

The most commonly used nuclear technique for the evaluation of myocardial ischemia/fibrosis in patients with DM is single-photon emission computerized tomography (SPECT), which is widely available; there is extensive experience of its use. However, SPECT has important limitations including the use of radiation and low spatial resolution that precludes detection of subendocardial ischemia/fibrosis. Furthermore, SPECT, because it is based on regional differences of myocardial blood flow, is less accurate in the detection of balanced ischemia, diffuse microvascular disease and subendocardial infarction [24]. A study of 575 patients with T2DM and angina, who underwent SPECT, found that myocardial ischemia was related to twice the risk of CV death or non-fatal myocardial infarction over a 4.4-year follow-up [25].

Nuclear techniques can detect CFR changes, and similarly to echocardiography, both symptomatic and asymptomatic patients with T2DM and normal coronary arteries may exhibit reduced CFR compared to controls, when assessed via technetium sestamibi scanning [26].

In contrast to SPECT, positron emission tomography (PET) uses less radiation and can be of great value regarding the evaluation of myocardial ischemia in DM. Myocardial perfusion has allowed incremental risk prediction of cardiac death in large cohorts of patients with or without DM, referred for clinical ^82^Rb stress PET [27]. However, it uses radiation, is less widely available, is expensive, and there is less expertise in the evaluation of its results. Finally, the combination of PET/computed tomography (CT) in T1DM patients showed comparable myocardial microvascular function between patients with T1DM and normoalbuminuria and non-diabetic controls, but impaired function if microvascular complications (macro-albuminuria or proliferative retinopathy) were present, while coronary calcification was increased in patients with DM, but was not explained by albuminuria [28].

PET can more accurately quantify CFR and myocardial blood flow (MBF). PET has shown significant alterations in the MBF of asymptomatic patients with T2DM (vs. controls) during dipyridamole stress [29]. There are also prognostic data using PET showing that patients with T2DM and impaired CFR had a prognosis similar to patients with CAD but without T2DM [30].

## 4. Cardiac Computed Tomography (CCT)

Cardiac computed tomography (CCT) allows non-invasive visualization of coronary arteries. Significant obstructive CAD is related to adverse cardiovascular outcomes, as in non-T2DM subjects. Regarding asymptomatic individuals, a meta-analysis of 10 studies pertaining to 5012 individuals with T2DM demonstrated that both non-obstructive and obstructive (≥50% stenosis) CAD were significantly associated with adverse events, as was the extent of CAD [31]. Patients with T2DM have higher calcium scores than non-T2DM subjects, and this is also associated with unfavorable prognosis [32].

Recently, it has been shown that in DM patients with stable chest pain, a computed tomographic coronary angiography (CTCA) approach was associated with fewer adverse CV events than a functional stress-testing strategy and that, therefore, it could be an initial diagnostic tool for these patients [33]. Furthermore, epicardial adipose tissue (EAT), as an inflammation marker, and plaque characteristics, denoting the extent of atherosclerotic CAD, are increased in DM and are related to adverse CV events [34]. Lastly, in T2DM patients undergoing clinically indicated CTCA, the identification of high LAD-pericoronary adipose tissue (PCAT) attenuation predicts CV events and identifies high-risk T2DM [35]. However, CTCA has some serious limitations, including the use of radiation and nephrotoxic contrast agents, high cost and low expertise. However, a meta-analysis of five randomized controlled trials (RCTs) including 3299 asymptomatic individuals with DM, demonstrated that non-invasive imaging for CAD did not significantly reduce non-fatal MIs and hospitalization for heart failure (HF) [36].

## 5. How Can Cardiovascular Magnetic Resonance (CMR) Imaging Be Used as a Useful Adjunct in the Cardiovascular Assessment of DM and PRE-DM Patients?

DM and PRE-DM are chronic diseases involving mainly the CV system and are considered as an equivalent to CAD. Therefore, there is a great clinical need to have a reliable diagnostic tool with good reproducibility, which can detect early myocardial alterations and can be applied repetitively without using radiation. CMR presents all these characteristics, because it has the best spatial resolution allowing it to detect small myocardial changes, missed by other imaging modalities. Additionally, CMR is the only modality that can evaluate cardiac function, stress-rest perfusion and perform tissue characterization [24].

CMR uses different types of images (sequences) to answer various questions regarding tissue characterization, perfusion and function in DM patients. The most common sequences in clinical practice include:**(i)** **Evaluation of Cardiac Function**

The CMR pulse sequence for functional evaluation is called balanced steady-state free precession (bSSFP) and represents the gold standard for the assessment of cardiac mass, wall motion, anatomy, as well as atrial and atrial/ventricular function [24].

**(ii)** 
**Evaluation of Myocardial Perfusion with Stress CMR**


Rapid cardiac imaging with the use of T1-W after pharmacologic hyperaemic stress with adenosine (or dipyridamole, ATP, regadenosine) and injection (bolus) of paramagnetic Gd-based contrast agent can provide accurate and reproducible data regarding myocardial perfusion during stress [18]. This technique allows the detection of perfusion flaws, due to micro- [18] or macro- vascular coronary artery disease [24].

Stress CMR imaging has no limitations depending on the body habitus, and is the method of choice for epicardial and micro-vascular CAD, especially for individuals who cannot exercise adequately or completely [24]. The CE-MARC [37] is the largest clinical trial of head-to-head comparisons between stress CMR and SPECT, that used coronary angiography as the reference standard and demonstrated the clinical value of stress CMR and its superiority over SPECT.

**(iii)** 
**Tissue Characterization**


This can be achieved using different types of images that reflect different cellular properties of myocardium. These images include:**(a)** **T1-weighted (T1-W) Images and Late Gadolinium Enhancement (LGE)**

T1-W imaging is the ideal sequence for CMR morphological assessment. Late gadolinium enhanced T1-W images (LGE), taken with the use of inversion recovery pulse sequences 10–15 min post gadolinium-based contrast administrations, represent the gold standard for the assessment and quantification of myocardial replacement fibrosis (scar) [37]. LGE can also detect significant extracellular interstitial expansion in cases of pulmonary hypertension, amyloidosis, or myocarditis [38].

The typical CMR image of myocardial infarction, commonly found in DM and PRE-DM, includes transmural or subendocardial LGE following the epicardial coronary arteries’ distribution. Subepicardial, patchy or intramyocardial LGE, usually in the inferolateral wall, is related to myocarditis, irrespective of etiology (infective, autoimmune, etc.). Furthermore, diffuse subendocardial LGE not following the typical distribution of coronary arteries’ territories is often related to microvascular coronary artery disease, as in DM with normal epicardial coronary arteries. Figure 1 presents characteristic CMR perfusion/fibrosis images in DM.

**(b)** 
**T1 Mapping and Extracellular Volume Fraction (ECV)**


LGE is the method of choice for detection and quantification of replacement fibrosis; however, it has inherent limitations to evaluate diffuse myocardial fibrosis, because it uses signal intensity differences between normal and scarred myocardium [38]. Parametric imaging, including T1 mapping and ECV, was developed to overcome this disadvantage. T1 mapping (native or pre-contrast T1 and post-contrast T1) enables the quantitative assessment of tissue T1 values and identifies diffuse myocardial fibrosis, which may not be detected by circulating biomarkers [38]. As field strength and different types of pulse sequences influence T1 measurements, it is recommended that different MRI units develop their own values that can be considered “normal” for use in their clinical practice [25]. Post-contrast T1 mapping is used for ECV calculation together with native T1 mapping. Measurement of myocardial and blood T1 pre- and post-administration of contrast agents is needed for estimation of the ECV, as follows:$$ECV=\left(1-Hematocrit\right)\times \frac{\left(1/T1_{\left(myo\;post\text{-}contrast\right)}\right)-\left(1/T1_{\left(myo\;pre\text{-}contrast\right)}\right)}{\left(1/T1_{\left(blood\;post\text{-}contrast\right)}-1/T1_{\left(blood\;pre\text{-}contrast\right)}\right)}$$

ECV is more reproducible than native and post-contrast T1 at different field strengths, acquisition methods and vendors, therefore showing better agreement with histology [38].

Native T1 mapping can increase in the remote myocardial infarction area, representing bad prognosis. Moreover, increased values of native T1 mapping and ECV can be early signs of various cardiomyopathies, before the detection of strain and strain rate abnormalities [38]. Furthermore, native T1 mapping is also sensitive to myocardial oedema, iron overload and diffuse scarring [38], allowing for the monitoring of longitudinal changes associated with treatment [38]. Both native T1 mapping and ECV have good correlation with endomyocardial biopsy regarding the fibrosis assessment [38]. Among patients with known or suspected CAD, T2D patients had a higher ECV and a history of T2D and high ECV were both independent risk factors for adverse CV outcomes [39].

**(c)** 
**T2-weighted (T2-W) Images**


T2-W images are the outcome of water accumulation, due to myocardial oedema [38], and reflect an acute myocardial response to the injury provoked by either inflammation (myocarditis) or ischemia (myocardial infarction). Myocardial injuries appear in T2W images as an area of high signal intensity on short tau inversion recovery (STIR T2) images, where the signal contrast between oedema, normal myocardium and LV cavity is optimized. Nevertheless, STIR T2 images have limitations, due to low signal to noise [38].

**(d)** 
**T2 mapping**


Has been proposed as a more reliable alternative to classic STIRT2 images. To overcome the problems of STIR T2, a parametric image of each voxel is reconstructed in T2 mapping. Although T2 measures are reproducible and independent of the heart rate and body habitus, they can vary with different field strengths or scanner types [38].

**(e)** 
**T2* mapping**


Is the most reliable index of the assessment of iron deposition and evaluation of chelation treatment in patients with iron overload, due to multiple blood transfusions as in thalassemia or myelodysplastic syndromes. Furthermore, it is an excellent index for assessing iron overload in patients with genetic iron metabolism disorders, such as hemochromatosis. However, more details on this topic are beyond the scope of this review [38].

**(iv)** 
**Magnetic Resonance Angiography**


Magnetic resonance angiography (MRA) is considered as accurate as X-ray angiography in the detection of great vessel anomalies, and an important tool for patients with DM [18]. Non-contrast angiography can produce important data concerning large vessel stenosis or aneurysms, and does not require administration of a contrast [38]. Contrast-enhanced MRA is commonly used to establish large vessel patency and to assess mural plaques in DM [38]. Finally, in a long-term follow-up, free-breathing whole heart coronary magnetic resonance angiography (CMRA) permits non-invasive risk stratification for major adverse cardiac events (MACE )and allows incremental prognostic value over conventional risk factors in patients without a history of coronary artery revascularization or myocardial infarction. Moreover, patients without obstructive CAD on CMRA have presented a good prognosis [40].

**(v)** 
**LV Deformation**


Clinically, a LVEF is commonly used to assess cardiac function. It should be noted that in the early stages of heart disease, a LVEF may be preserved despite alterations in the myocardial contractility. LV deformation expressed as a myocardial strain is a useful tool to evaluate the subclinical LV alterations. Myocardial strain can be measured by echocardiography, CMR tissue tracking and MR tagging. Echocardiography is frequently used for strain analysis, but its accuracy is dependent on operator training and patient body habitus. CMR tagging requires special sequences, whereas the post-processing assessment is time-consuming and complex. Lastly, CMR tissue tracking is considered a quantitative technique allowing fast analysis of the myocardial strain [41].

CMR tissue tracking can evaluate the motion of myocardial voxel and provide important data concerning myocardial strain in subclinical myocardial abnormalities before myocardial injury leads to significant decrease in the LVEF. SSFP cine sequences are appropriate for CMR tissue tracking, which, using a rapid post-processing system, may provide ideal evaluation of LV myocardial strain. In a recent study evaluating patients with T2DM, circumferential PS and global radial peak strain (PS) did not differ significantly between patients with a preserved LVEF and controls. Nevertheless, longitudinal PS was significantly lower in patients with a preserved LVEF than controls, suggesting an independent association of global longitudinal PS with the LV remodeling index [41].

**(vi)** 
**Magnetic Resonance Spectroscopy (MRS)**


Magnetic resonance spectroscopy (MRS), with the use of proton (^1^H), can detect increased myocardial triglyceride content in patients with T2DM, which reflects cardiac steatosis that can also be present in subjects with a preserved LVEF [42]. Steatosis is associated with concentric LV remodeling, reduction in the systolic strain, and impaired myocardial energetics detected by ^31^P (phosphorus) MRS [43]. Lastly, the myocardial phosphocreatinine to adenosine triphosphate ratio (PCr/ATP) measurement using ^31^P MRS represents a non-invasive index of myocardial energetics. This is diminished in patients with T2DM and correlates with coronary microvascular dysfunction, reduced myocardial oxygenation/perfusion, and diastolic or subclinical systolic dysfunction [44].

## 6. CMR Limitations

Although the expertise and availability of CMR are generally low, today there is an increase in CMR in clinical practice.CMR is contraindicated in patients with non-MRI conditional devices or metallic clips [38].Caution is needed for the use of gadolinium-based contrast agents (CA) in patients with renal impairment (GFR < 30 mL/min) [38]. It should be noted that several non-contrast sequences can provide important information, irrespective of the use of a contrast agent [38].Using CMR in the decision-making process has a clear benefit evident in a cost-minimization analysis for cardiac revascularization of these patients [45].

## 7. Can CMR Be a Useful Imaging Adjunct to ESC Guidelines for DM?

Recently, ESC published guidelines regarding the management of CVD in DM [11]. The recommendations were based on adequate glycemic control and modification of the existing risk factors for CVD. However, no details were provided on how to evaluate the status of myocardium per se. It is well documented that the metabolic abnormalities in DM can alter the function of cardiomyocytes before the development of cardiac symptoms and these alterations can be reliably detected using cardiac imaging methods.

Furthermore, in the assessment of CAD in patients with DM, the distinction between T1DM and T2DM regarding the pathophysiology of CAD should be taken into serious consideration. It is well documented that microvascular coronary artery disease (M-CAD) is the main type of CAD found in T1DM, while epicardial coronary artery disease (E-CAD) is more common in T2DM [28]. In this context, patients with T1DM should be regularly evaluated for M-CAD, which may remain asymptomatic or with atypical symptoms for many years [29].

Cardiovascular magnetic resonance (CMR) and positron emission tomography (PET), on the other hand, represent the ideal imaging methods for the assessment of M-CAD [46]. However, in contrast to CMR, PET is less available, is more expensive, and uses radiation. Furthermore, adenosine stress CMR can differentiate between M-CAD and E-CAD with X-ray coronary angiography being recommended only in patients with a prospect of invasive intervention. Furthermore, patients with DM demonstrate a blunted maximal non-contrast T1 response during adenosine vasodilatory stress, which reflects coronary microvascular dysfunction [47]. Therefore, adenosine stress/rest T1 mapping can be applied to patients with a renal dysfunction and/or known allergy to gadolinium-based contrast agents. As such, CMR can detect early diabetic heart phenotypic features in patients with T2DM, thus providing a chance for timely therapeutic intervention [48]. Lastly, CMR can clarify the pathophysiologic background to HF in DM (microvascular, subendocardial/transmural fibrosis and/or inflammation) and assess the potential presence of a silent myocardial infarction, which may be missed by other imaging modalities. Therefore, a suggested CV imaging approach for the evaluation of CAD in DM and PRE-DM should include:Routine baseline echocardiographic evaluation on an annual basis in parallel with evaluation of microalbuminuria, glycemic status and lipid profile of all DM patients.Evaluation of silent or overt perfusion/fibrosis lesions. In this context, CMR is preferable to other imaging modalities as having the best reproducibility and providing all the necessary information, missed by other imaging modalities, without using ionizing radiation. Furthermore, the CMR detection of silent CVD may seriously impact the DM patient treatment motivating the early use of ACE-inhibitors, b-blockers, ranolazine and other cardioprotective medications.

## 8. Conclusions

Diabetes mellitus (DM) is the new epidemic of our time, presenting a rapid increase worldwide. The recently published ESC guidelines focus on the accurate assessment of themetabolic profile and modification of CVD risk factors. However, CV imaging should be considered as a useful adjunct in the clinical evaluation of all patients with DM or PRE-DM.

Echocardiography should be recommended as the routine primary cardiac evaluation together with all required blood tests on the basis of annual evaluation for CAD. Stress echocardiography and SPECT, although they are the routinely used methods for fibrosis/ischemia assessment, both carry various limitations, such as acoustic window dependency and low spatial resolution, respectively. Furthermore, SPECT uses radiation, which does not allow its routine application on an annual basis. Therefore, for the evaluation of myocardial fibrosis/ischemia in DM, CMR seems clearly superior to all other imaging tools, as it is non-invasive without radiation and with high spatial resolution and excellent reproducibility that can detect small infarcts missed by other imaging modalities, can reliably differentiate M-CAD from E-CAD and guide further risk stratification/treatment evaluation of these patients.

## Figures and Tables

**Figure 1 jpm-13-00724-f001:**
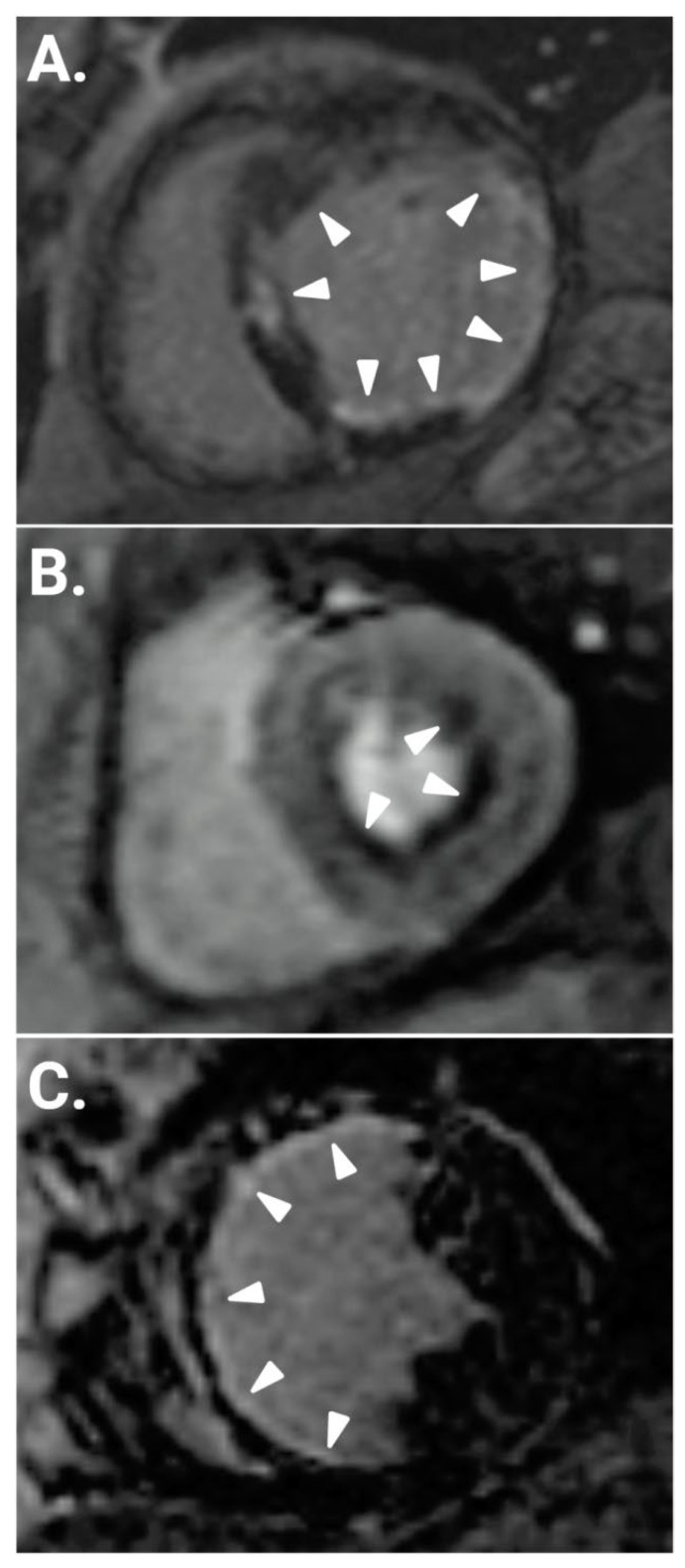
(**A**). Short axis inversion recovery CMR image showing silent subendocardial fibrosis (white arrowheads), due to M-CAD in a T1DM patient with normal epicardial coronary arteries, who was evaluated for heart failure. (**B**). Short axis first pass gadolinium T1 image showing silent, diffuse subendocardial perfusion defects (white arrowheads), due to M-CAD in a T1DM patient with normal epicardial coronary arteries, who presented with atypical chest pain. (**C**). Short axis inversion recovery CMR image showing extensive, subendocardial myocardial infarction, secondary to occlusion (white arrowheads) of the left anterior descending artery (LAD) in a patient with T2DM who was evaluated for heart failure. CMR: cardiovascular magnetic resonance; M-CAD: microvascular coronary artery disease; T1DM: Type 1 diabetes mellitus; T2DM: Type 2 diabetes mellitus.

## Data Availability

Not applicable.

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
