# Peer review of "Cardiovascular Magnetic Resonance Imaging as an Adjunct to the Evaluation of Cardiovascular Involvement in Diabetes Mellitus"

_jpm, 2023, doi:10.3390/jpm13050724_

Round 1

Reviewer 1 Report

The topic analysed by the authors is of important interest, in my opinion. as they also say, cardiovascular imaging is currently a 'must' in clinical cardiovascular assessment. Alterations in cardiovascular imaging parameters can lead to the early identification of various cardiovascular diseases, and I therefore find this work extremely relevant.

Author Response

We would like to thank the Editor and the Reviewers for giving us the opportunity to submit a revised version of our manuscript. We appreciate the Reviewers’ insight and we hope that we accommodated most of the suggestions in the revised manuscript. All changes have been tracked in the revised manuscript.

Reviewer 1

The topic analysed by the authors is of important interest, in my opinion. as they also say, cardiovascular imaging is currently a 'must' in clinical cardiovascular assessment. Alterations in cardiovascular imaging parameters can lead to the early identification of various cardiovascular diseases, and I therefore find this work extremely relevant.

We thank the Reviewer for these comments.

Reviewer 2 Report

The review done by Mavrogeni SI and cols. entitled: “Cardiovascular Magnetic Resonance Imaging as adjunct to the evaluation of Cardiovascular Involvement in Diabetes Mellitus” describes a series of medical tools (i.e. imaging sources) focused mainly on cardiovascular disease and T2 diabetes related CVD, since in T1D there is a lot of  reports indicating the successful useful of imaging there is a restricted knowledge on T2D. The authors stated the relevance of implementing the use of imaging tools for the effective prevention of cardiovascular diseases in T2D patients. Authors emphasized on the most accurate time point and how to make such imaging evaluations in these patients, and therefore imaging use may lead to the early identification of different types of CVD.

Authors deeply describe a wide number of different modalities of imaging tools and highlighting the minimum use of radiation and non-invasive characteristics of imaging tools.

Although, the review is well done and written it is lacking of few relevant points listed below:

1.       Abbreviations are not fully described in the manuscript

2.       In the chapter entitled “Tissue characterization” there is a poor description of the methods and variants implemented on T2 and T2* mapping   

3.       Conclusion is poorly described, it would be appreciate if authors go depth in to the advantages of the different modalities of  imaging highlighting the most recent tools applied for the detection of cardiac abnormalities

Author Response

We would like to thank the Editor and the Reviewers for giving us the opportunity to submit a revised version of our manuscript. We appreciate the Reviewers’ insight and we hope that we accommodated most of the suggestions in the revised manuscript. All changes have been tracked in the revised manuscript.

Reviewer 2

The review done by Mavrogeni SI and cols. entitled: “Cardiovascular Magnetic Resonance Imaging as adjunct to the evaluation of Cardiovascular Involvement in Diabetes Mellitus” describes a series of medical tools (i.e. imaging sources) focused mainly on cardiovascular disease and T2 diabetes related CVD, since in T1D there is a lot of  reports indicating the successful useful of imaging there is a restricted knowledge on T2D. The authors stated the relevance of implementing the use of imaging tools for the effective prevention of cardiovascular diseases in T2D patients. Authors emphasized on the most accurate time point and how to make such imaging evaluations in these patients, and therefore imaging use may lead to the early identification of different types of CVD.

Authors deeply describe a wide number of different modalities of imaging tools and highlighting the minimum use of radiation and non-invasive characteristics of imaging tools.

Although, the review is well done and written it is lacking of few relevant points listed below:

Comment 1

      Abbreviations are not fully described in the manuscript

      Answer

      Abbreviations are now fully described in the manuscript

Comment 2

       In the chapter entitled “Tissue characterization” there is a poor description of the methods

and variants implemented on T2 and T2* mapping   

       Answer

     The paragraphs on T2 and T2* mapping have been revised. 

Comment 3

Conclusion is poorly described, it would be appreciate if authors go depth in to the advantages of the different modalities of  imaging highlighting the most recent tools applied for the detection of cardiac abnormalities

Answer

Conclusion has been revised accordingly.